# Safety, effectiveness and immunogenicity of heterologous mRNA-1273 boost after prime with Ad26.COV2.S among healthcare workers in South Africa: The single-arm, open-label, phase 3 SHERPA study

Nigel Garrett[1,2]*, Tarylee Reddy[3], Nonhlanhla Yende-Zuma[1,3], Azwidhwi Takalani[4], Kubashni Woeber[5], Annie Bodenstein[6], Phumeza Jonas[6], Imke Engelbrecht[6], Waasila Jassat[7], Harry Moultrie[7], Debbie Bradshaw[8], Ishen Seocharan[3], Jackline Odhiambo[4], Kentse Khuto[4], Simone I. Richardson[9,10], Millicent A. Omondi[11], Rofhiwa Nesamari[11], Roanne S. Keeton[11], Catherine Riou[11,12], Thandeka Moyo-Gwete[9,10], Craig Innes[13], Zwelethu Zwane[13], Kathy Mngadi[13], William Brumskine[13], Nivashnee Naicker[1], Disebo Potloane[1], Sharlaa Badal-Faesen[14], Steve Innes[15], Shaun Barnabas[16], Johan Lombaard[17], Katherine Gill[15], Maphoshane Nchabeleng[18], Elizma Snyman[19], Friedrich Petrick[20], Elizabeth Spooner[21], Logashvari Naidoo[21], Dishiki Kalonji[21], Vimla Naicker[21], Nishanta Singh[21], Rebone Maboa[22], Pamela Mda[23], Daniel Malan[24], Anusha Nana[25], Mookho Malahleha[26], Philip Kotze[27], Jon J. Allagappen[28], Andreas H. Diacon[29], Gertruida M. Kruger[30], Faeezah Patel[31], Penny L. Moore[1,9,10], Wendy A. Burgers[11,12], Kate Anteyi[32], Brett Leav[32], Linda-Gail Bekker[15], Glenda E. Gray[33], Ameena Goga[34,35], the SHERPA study team[¶]

1 Centre for the AIDS Programme of Research in South Africa, Durban, South Africa, 2 Discipline of Public Health Medicine, School of Nursing and Public Health, University of KwaZulu-Natal, Durban, South Africa, 3 Biostatistics Research Unit, South African Medical Research Council (SAMRC), Durban, South Africa, 4 Hutchinson Center Research Institute of South Africa, Cape Town, South Africa, 5 Grants, Innovation and Product Development Unit, SAMRC, Durban, South Africa, 6 Right to Care, Johannesburg, Gauteng, South Africa, 7 Division of Public Health Surveillance and Response, National Institute for Communicable Diseases (NICD) of the National Health Laboratory Services (NHLS), Johannesburg, Gauteng, South Africa, 8 Burden of Disease Research Unit, SAMRC, Tygerberg, South Africa, 9 SAMRC Antibody Immunity Research Unit, School of Pathology, University of the Witwatersrand, Johannesburg, Gauteng, South Africa, 10 Centre for HIV and STIs, NICD of the NHLS, Johannesburg, Gauteng, South Africa, 11 Institute of Infectious Disease and Molecular Medicine, Division of Medical Virology, Department of Pathology, University of Cape Town, Cape Town, South Africa, 12 Wellcome Centre for Infectious Diseases Research in Africa, University of Cape Town, Cape Town, South Africa, 13 Aurum Institute, Johannesburg, Gauteng, South Africa, 14 Clinical HIV Research Unit, Johannesburg, Gauteng, South Africa, 15 Desmond Tutu HIV Centre, University of Cape Town, Cape Town, South Africa, 16 Family Centre for Research with Ubuntu, Tygerberg, South Africa, 17 Josha Research, Bloemfontein, South Africa, 18 Medunsa Clinical Research Unit, Sefako Makgatho Health Sciences University, Pretoria, South Africa, 19 TASK Eden, George, South Africa, 20 Mzansi Ethical Research Centre, Middelburg, South Africa, 21 SAMRC Clinical Research Site, Durban South Africa, 22 Ndlovu Research Centre, Elandsdoorn, South Africa, 23 Nelson Mandela Academic Research Unit CRS, Mthatha, South Africa, 24 PHOENIX Pharma, Port Elizabeth, South Africa, 25 Perinatal HIV Research Unit, Johannesburg, Gauteng, South Africa, 26 Synergy Biomed Research Institute, East London, South Africa, 27 Qhakaza Mbokodo Research Clinic, Ladysmith, South Africa, 28 Setshaba Research Centre, Soshanguve, South Africa, 29 TASK Central, Cape Town, South Africa, 30 Ubuntu Clinical Research, Krugersdorp, Gauteng, South Africa, 31 Wits RHI Research Institute, University of Witwatersrand, Johannesburg, Gauteng, South Africa, 32 Moderna Inc., Cambridge, Massachusetts, United States of America, 33 Office of the President and CEO, SAMRC, Cape Town, South Africa, 34 Department of Paediatrics and Child Health, University of Pretoria, Pretoria, South Africa, 35 HIV and other Infectious Diseases Research Unit, SAMRC, Durban, South Africa

¶ Membership of the SHERPA study team is listed in the Acknowledgments.
* nigel.garrett@caprisa.org



**Data Availability Statement:** You can access the SHERPA data on the following link: https://medat.samrc.ac.za/index.php/catalog/56.

**Funding:** The SHERPA study was funded by Moderna, Inc. (Cambridge, Massachusetts, US) and the South African Medical Research Council (SAMRC). Moderna provided mRNA-1273 free-of-charge. The Sisonke trial was funded by: The National Department of Health through baseline funding to the SAMRC; the Solidarity Response Fund NPC; The Michael & Susan Dell Foundation; the ELMA Vaccines and Immunization Foundation (21-V0001, G.E.G.); and the Bill & Melinda Gates Foundation (INV-030342, G.E.G.). Moderna representatives reviewed the study protocol, participated in safety oversight, and contributed as manuscript co-authors, but were not involved in data collection and analysis.

**Competing interests:** KA and BL are employees of Moderna, Inc. and may hold stock/stock options in the company. The other authors declare no conflict of interests.

## Abstract

Limited studies have been conducted on the safety and effectiveness of heterologous COVID-19 vaccine boosting in lower income settings, especially those with high-HIV prevalence., The Sisonke Heterologous mRNA-1273 boost after prime with Ad26.COV2.S (SHERPA) trial evaluated a mRNA-1273 boost after Ad26.COV2.S priming in South Africa. SHERPA was a single-arm, open-label, phase 3 study nested in the Sisonke implementation trial of 500000 healthcare workers (HCWs). Sisonke participants were offered mRNA-1273 boosters between May and November 2022, when Omicron sub-lineages were circulating. Adverse events (AE) were self-reported, and co-primary endpoints (SARS-CoV-2 infections and COVID-19 hospitalizations or deaths) were collected through national databases. We used Cox regression models with booster status as a time-varying covariate to determine the relative vaccine effectiveness (rVE) of the mRNA-1273 booster among SHERPA versus unboosted Sisonke participants. Of 11248 SHERPA participants in the rVE analysis cohort (79.3% female, median age 41), 45.4% had received one and 54.6% two Ad26.COV2.S doses. Self-reported comorbidities included HIV (18.7%), hypertension (12.9%) and diabetes (4.6%). In multivariable analysis including 413161 unboosted Sisonke participants, rVE of the booster was 59% (95%CI 29–76%) against SARS-CoV-2 infection: 77% (95%CI 9–94%) in the one-Ad26.COV2.S dose group and 52% (95%CI 13–73%) in the two-dose group. Severe COVID-19 was identified in 148 unboosted Sisonke participants, and only one SHERPA participant with severe HIV-related immunosuppression. Of 11798 participants in the safety analysis, 228 (1.9%) participants reported 575 reactogenicity events within 7 days of the booster (most commonly injection site pain, malaise, myalgia, swelling, induration and fever). More reactogenicity events were reported among those with prior SARS-CoV-2 infections (adjusted odds ratio [aOR] 2.03, 95%CI 1.59–2.59) and less among people living with HIV (PLWH) (aOR 0.49, 95%CI 0.34–0.69). There were 115 unsolicited adverse events (AEs) within 28 days of vaccination. No related serious AEs were reported. In an immunogenicity sub-study, mRNA-1273 increased binding and neutralizing antibody titres and spike-specific T-cell responses 4 weeks after boosting regardless of the number of prior Ad26.COV2.S doses, or HIV status, and generated Omicron spike-specific cross-reactive responses. mRNA-1273 boosters after one or two Ad26.COV2.S doses were well-tolerated, safe and effective against Omicron SARS-CoV-2 infections among HCWs and PLWH.

**Trial registration:** The SHERPA study is registered in the Pan African Clinical Trials Registry (PACTR): PACTR202310615330649 and the South African National Clinical Trial Registry (SANCTR): DOH-27-052022-5778.

## Introduction

During the first 2 years of the COVID-19 pandemic (2020–2022), populations in low- and middle-income countries (LMICs) had limited access to mRNA vaccines. Most LMICs including South Africa, started their COVID-19 vaccination programmes with vector-based vaccines, such as Ad26.COV2.S (Janssen/Johnson & Johnson) and ChAdOx1-S (Oxford/AstraZeneca). When vaccine demand subsided in high-income countries and production

increased, mRNA vaccines became available, allowing for a heterologous boosting strategy in LMICs. While this strategy was supported by immunogenicity data and by programmatic evaluations [1, 2], it was not formally investigated in clinical trials, meaning there was limited evidence on safety and effectiveness of mRNA booster vaccinations after the Ad26.COV2.S vaccine, especially from LMICs with high HIV prevalence.

In South Africa, the COVID-19 vaccination rollout started in February 2021, when half a million healthcare workers received a single dose of Ad26.COV2.S in the Sisonke phase 3b implementation trial [3]. This occurred just before the rapid spread of the SARS-CoV-2 Delta variant. Sisonke showed that a single dose of Ad26.COV2.S was 67% (95% confidence interval [CI] 62–71%) effective against COVID-19-related hospitalisations and 83% (75–89%) against COVID-19-related deaths [3]. Following the release of the ENSEMBLE 2 trial results [4, 5], approximately half of the Sisonke participants opted for a second Ad26.COV2.S dose between October and December 2021. By this time, South Africa experienced a surge in SARS-CoV-2 cases with the Omicron variant. Early data showed that the second Ad26.COV2.S dose provided added protection against hospitalizations and deaths during the Omicron wave [6].

Following the recommendation by the United States Food and Drug Administration to limit the use of the Ad26.COV2.S vaccine, because of rare events of thrombosis with thrombocytopenia syndrome, and emerging data showing superior neutralizing antibody responses after mRNA vaccines, there was an urgent need to clarify the safety, immunogenicity and effectiveness of a heterologous vaccination approach in LMICs. Considering this, the Sisonke study team rapidly implemented and conducted the SHERPA study nested in Sisonke to investigate a heterologous mRNA-1273 boost after one or two doses of Ad26.COV2.S among healthcare workers in South Africa.

## Methods and materials

### Study design and setting

SHERPA (Sisonke Heterologous mRNA-1273 boost after Prime with Ad26.COV2.S) was a single-arm, open-label, phase 3 study, nested in the Sisonke trial, in which healthcare workers in South Africa had received one or two doses of Ad26.COV2.S between February and December 2021. In SHERPA, Sisonke participants were offered a mRNA-1273 booster dose (ancestral strain) at 30 clinical research sites (CRS) across South Africa, between 23 May 2022 and 12 November 2022. Using routinely collected national data, we aimed to compare SARS-CoV-2 infections and COVID-19-related hospitalizations and deaths among SHERPA participants with Sisonke participants who did not receive the mRNA-1273 booster (see study schema in **S1 Table**). The data cut-off for outcome ascertainment was 31 December 2022. Seven CRS followed 200 SHERPA participants for six months in a safety and immunogenicity sub-study. SHERPA was registered in the Pan African Clinical Trials Registry (PACTR202310615330649) and the South African National Clinical Trial Registry (DOH-27-052022-5778). All participants provided written informed consent.

### Participants

Sisonke participants were invited to join SHERPA via short messaging system (SMS) with a study website link, where they could register and choose a convenient CRS. SHERPA participants were 18 years or older with or without comorbidities, including HIV infection, were employed in the South African public or private healthcare sector, and had received either one or two doses of Ad26.COV2.S in Sisonke. Considering the high COVID-19 morbidity among pregnant women, pregnant and breastfeeding women were eligible. Sisonke participants who had received any COVID-19 vaccine other than one or two doses of Ad26.COV2.S (e.g.,

another mRNA booster) and those with evidence of SARS-CoV-2 infection within 14 days of enrolment were excluded (see **S2 Table** for full eligibility criteria).

## Study procedures

Once consented, the study team took a medical history, conducted a physical examination, and recorded any concomitant medications. Female participants of reproductive age had a pregnancy test, and all participants had a nasal swab taken for SARS-CoV-2 polymerase chain reaction (PCR) testing and a plasma sample to test for SARS-CoV-2 serology (Architect, Abbott, Chicago, IL, USA) before vaccination. Participants in the safety and immunogenicity sub-study also had peripheral blood mononuclear cells (PBMC) and plasma samples taken before mRNA-1273 booster vaccination and 4 and 24 weeks after vaccination (see **S3** and **S4** **Tables**).

The 50 microgram dose of the mRNA-1273 booster was prepared by research pharmacists. Clinical research staff then administered the vaccine in the deltoid muscle, and participants were monitored for 15 minutes for immediate reactogenicity events or allergic reactions. Adverse events (AE) experienced at the vaccination visit were managed and recorded by the CRS team. After the observation period, participants were issued with vaccination certificates, and CRS and Sisonke 'Safety Desk' contact details for AE reporting. Pregnant and breastfeeding women were followed up by the central safety team, through a pregnancy and breastfeeding registry that both the CRS and safety teams had access to.

## Immunogenicity sub-study

Plasma and PBMC samples were taken before mRNA-1273 vaccination and 4 and 24 weeks after vaccination. We conducted validated SARS-CoV-2 spike enzyme linked immunosorbent assays (ELISA), SARS-CoV-2 nucleocapsid ELISA testing, a lentiviral pseudovirus neutralization assay, and an antibody-dependent cellular cytotoxicity (ADCC) assay at all timepoints. Neutralizing antibody responses in people living with HIV (PLWH) were tested using a vesicular stomatitis virus-based neutralization assay which was not subject to antiretroviral therapy interference [7], and which produced equivalent titers to the lentiviral assay (**S1 Fig**). T-cell responses to SARS-CoV-2 spike were measured from cryopreserved PBMC using intracellular cytokine staining and flow cytometry. This was performed on the first 50 participants with available PBMC samples at the three sampling visits (see further details on immunogenicity assays in **S1 Text**).

## Data sources and data management

We obtained SARS-CoV-2 and severity endpoint data through the surveillance system that was built for Sisonke (**S2 Fig**). Data sources included the Notifiable Medical Conditions Sentinel Surveillance (NMCSS) database of SARS-CoV-2 cases, the DATCOV database of reported confirmed COVID-19 hospital admissions, and the national Electronic Vaccination Data System (EVDS) for COVID-19 vaccines. Automated near real-time deterministic and probabilistic linkage algorithms were applied to link these three routine COVID-19 datasets. Breakthrough SARS-CoV-2 infections, defined as a positive SARS-CoV-2 PCR or antigen test ≥14 days after booster vaccination, and COVID-19 hospitalisations or deaths were identified through linking the databases.

SHERPA study visits were recorded in a Research Electronic Data Capture (REDCap) database (Vanderbilt University, Nashville, US). After the screening/enrolment visit, solicited and unsolicited AEs were self-reported via an online data entry link shared by SMS with participants 1, 7 and 28 days after the booster. All safety events including Serious AEs (SAE) and AEs

of Special Interest (AESI) were reviewed by safety clinicians and the Protocol Safety Review Team (PSRT). Data management systems were CFR Part 11 and POPIA compliant and processes followed Good Clinical Data Management Processes guidance.

## Variables

For the relative vaccine effectiveness (rVE) analysis, we used demographic and clinical variables that are routinely collected in the EVDS, including age, sex, common comorbidities, and geographic location. The NMCSS database provided information about previous SARS-CoV-2 infections.

## Outcomes

The co-primary outcomes were to assess the rVE of the mRNA-1273 booster by comparing the rates of SARS-CoV-2 infections and severe COVID-19 cases (defined as COVID-19 hospitalizations or deaths) among SHERPA participants compared to unboosted Sisonke participants. Secondary outcomes included an assessment of rVE of the mRNA-1273 booster against SARS-CoV-2 Infections, COVID-19 hospitalizations and deaths, in those who had received one or two doses of Ad26.COV2.S separately; an assessment of safety of the mRNA-1273 boost, including among pregnant women and PLWH; and comparisons of antibody and T-cell responses before and after vaccination.

## Statistical analysis

Analyses for the primary and secondary outcomes were performed using SAS version 9.4 (Statistical Analysis Software, North Carolina, USA) and Stata version 17 (College Station, TX, USA). Demographic and clinical data of participants enrolled in SHERPA and Sisonke were summarized using descriptive statistics including frequencies, proportions, and medians with interquartile ranges (IQR). For the primary analysis, we used a statistical method proposed by Fintzi and Follmann [8] referred to here as an adjusted cohort approach, which allowed inclusion of SHERPA participants within the larger Sisonke cohort (S3 Fig).

We used a Cox regression model with the mRNA-1273 booster status as a time-varying covariate, adjusting for age, sex, an indicator of whether a participant had previously received one or two doses of Ad26.COV2.S, geographic location, number of comorbidities and evidence of prior SARS-CoV-2 infection. This time-varying exposure was characterized by a continuous piecewise linear function of time elapsed since the first mRNA-1273 dose (23 May 2022) for the log hazard ratio. Participants who did not experience an event of interest were censored on 31 December 2022 (cut-off date). The rVE in reducing the risk of SARS-CoV-2 infection and COVID-19 hospitalizations and deaths was estimated as one minus the estimated hazard ratio with 95% CI. Further details regarding outcome analyses, including an alternative matched cohort analysis approach, definitions of variables, and immunogenicity analyses are detailed in the **Statistical Analysis Plan**.

## Study approvals

The SHERPA and Sisonke trial protocols were approved by the South African Health Products Regulatory Authority (SAHPRA; 20210463), and health research ethics committees of all participating CRSs. The study received full ethical approval by the following ethics committees in South Africa: Pharma-Ethics (220324629), Stellenbosch University Health Research Ethics Committee (M22/03/003_COVID-19), University of KwaZulu-Natal Biomedical Research Ethics Committee (BREC/00004119/2022), University of Cape Town Human Research Ethics

Committee (301/2022), Sefako Makgatho University Research Ethics Committee (SMUREC/ M/85/2022), The South African Medical Research Council Human Research Ethics Committee (EC007-3/2022) and the University of the Witwatersrand Human Research Ethics Committee (220402B).

## Results

### Study population

Of 501230 Sisonke participants who had received one or two Ad26.COV2.S doses between 17 February 2021 and 22 May 2022, 12342 enrolled into SHERPA and received the mRNA-1273 booster between 23 May 2022 and 12 November 2022 (**Fig 1**). Of these, 1094 (8.9%) were excluded from the rVE analysis, because they had either received a third Ad26.COV2.S or a BNT162b2 mRNA booster, or had received an Ad26.COV2.S vaccine after the earliest date of mRNA-1273 vaccination on 23 May 2022. Participants at one CRS had to be excluded from rVE and safety analyses due to concerns about data integrity. The mRNA-1273 booster rVE analysis included 11248 SHERPA participants, of whom 5102 (45.4%) had received one and 6146 (54.6%) two prior Ad26.COV2.S doses. These were compared to 413161 non-SHERPA Sisonke participants, of whom 185676 (44.9%) had received one and 227485 (55.1%) two doses of Ad26.COV2.S.

Baseline demographic and clinical characteristics of the SHERPA and non-SHERPA participants stratified by number of previous Ad26.COV2.S vaccinations are summarized in **Table 1**. The majority (79.3%) of SHERPA participants were female and 20.7% were male. The median age was 41 years (IQR 35–48). One third (32.2%) of participants reported at least one comorbidity, most commonly HIV infection (18.7%), hypertension (12.9%) and diabetes (4.6%). Participants were recruited from all nine South African provinces. Non-SHERPA participants had similar characteristics, apart from fewer reporting HIV infection (8.3%).

### Prior SARS-CoV-2 infections and time since last vaccination

Participants were checked against the NMCSS database for evidence of previous SARS-CoV-2 infection. Among SHERPA participants 1079 (9.2%) had evidence of a previous SARS-CoV-2 infection compared to 54770 (13.3%) among the non-SHERPA group. Overall, these estimates were likely an underestimate considering higher self-report rates (28%) and evidence of high

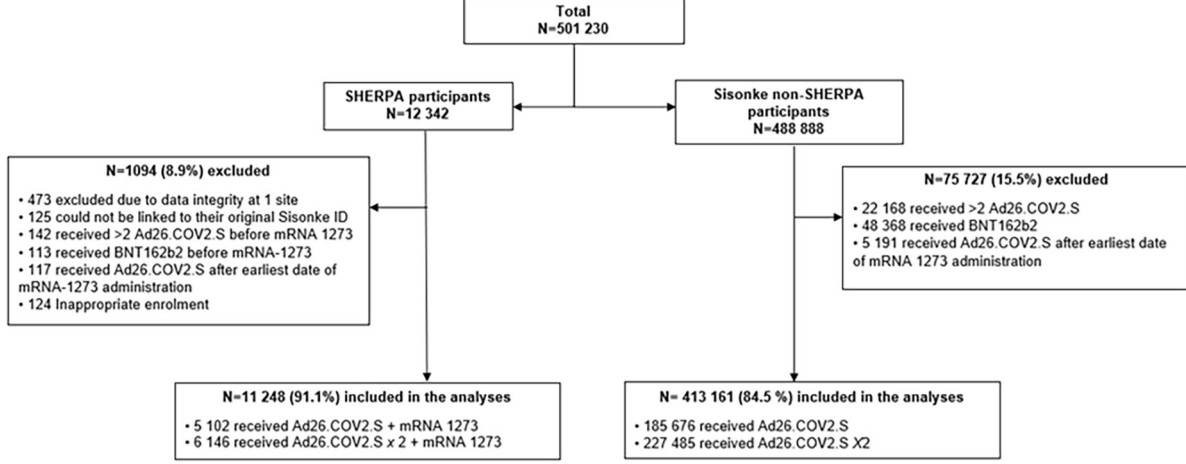

**Fig 1. Flow chart of the SHERPA analysis cohort nested in the Sisonke implementation trial.**

**Table 1. Baseline demographics and clinical characteristics of SHERPA and non-SHERPA participants previously vaccinated in Sisonke.**

| Characteristic | SHERPA | | | non-SHERPA | | |
|---|---|---|---|---|---|---|
| | Ad26.COV2.S + mRNA-1273 (N = 5102) | 2 Ad26.COV2.S + mRNA-1273 (N = 6146) | Total (N = 11248) | Ad26.COV2.S (N = 185676) | 2 Ad26.COV2.S (N = 227485) | Total (N = 413161) |
| **Sex, n (%)** | | | | | | |
| Female | 4048 (79.3%) | 4877 (79.4%) | 8925 (79.3%) | 136534 (73.5%) | 171854 (75.5%) | 308388 (74.6%) |
| Male | 1054 (20.7%) | 1269 (20.6%) | 2323 (20.7%) | 49142 (26.5%) | 55631 (24.5%) | 104773 (25.4%) |
| Median age (years), IQR | 40 (33–46) | 43 (36–50) | 41 (35–48) | 40 (33–49) | 44 (35–53) | 42 (34–51) |
| **Age groups (year), n (%)** | | | | | | |
| 18–39 | 2538 (49.7%) | 2297 (37.4%) | 4835 (43.0%) | 90048 (48.5%) | 83720 (36.8%) | 173768 (42.1%) |
| 40–49 | 1743 (34.2%) | 2288 (37.2%) | 4031 (35.8%) | 53390 (28.8%) | 68471 (30.1%) | 121861 (29.5%) |
| 50–59 | 699 (13.7%) | 1257 (20.5%) | 1956 (17.4%) | 31549 (17.0%) | 53257 (23.4%) | 84806 (20.5%) |
| $\geq$60 | 122 (2.4%) | 304 (4.9%) | 426 (3.8%) | 10628 (5.7%) | 22034 (9.7%) | 32662 (7.9%) |
| Missing | 0 | 0 | 0 | 61 (<0.1%) | 3 (<0.1%) | 64 (<0.1%) |
| **Number of comorbidities, n (%)** | | | | | | |
| 0 | 3573 (70.0%) | 4048 (65.9%) | 7621 (67.8%) | 141683 (76.3%) | 166384 (73.1%) | 308067 (74.6%) |
| 1 | 1334 (26.1%) | 1792 (29.2%) | 3126 (27.8%) | 37390 (20.1%) | 50328 (22.1%) | 87718 (21.2%) |
| $\geq$2 | 195 (3.8%) | 304 (4.9%) | 501 (4.5%) | 6603 (3.6%) | 10773 (4.7%) | 17376 (4.2%) |
| **Risk factors for severe COVID-19, n (%)** | | | | | | |
| Living with HIV | 977 (19.1%) | 1122 (18.3%) | 2099 (18.7%) | 17612 (9.5%) | 16572 (7.3%) | 34184 (8.3%) |
| Hypertension | 541 (10.6%) | 912 (14.8%) | 1453 (12.9%) | 23007 (12.4%) | 38094 (16.7%) | 61101 (14.8%) |
| Diabetes | 188 (3.7%) | 332 (5.4%) | 520 (4.6%) | 8382 (4.5%) | 14471 (6.4%) | 22853 (5.5%) |
| Cancer | 5 (0.1%) | 15 (0.2%) | 20 (0.2%) | 355 (0.2%) | 680 (0.3%) | 1035 (0.3%) |
| Tuberculosis | 10 (0.2%) | 12 (0.2%) | 22 (0.2%) | 236 (0.1%) | 209 (0.1%) | 445 (0.1%) |
| Heart disease | 9 (0.2%) | 26 (0.4%) | 35 (0.3%) | 974 (0.5%) | 1661 (0.7%) | 2635 (0.6%) |
| Chronic lung disease | 5 (0.1%) | 14 (0.2%) | 19 (0.2%) | 481 (0.3%) | 901 (0.4%) | 1382 (0.3%) |
| **Geographical location, n (%)** | | | | | | |
| Eastern Cape | 762 (14.9%) | 838 (13.6%) | 1600 (14.2%) | 25454 (13.7%) | 24773 (10.9%) | 50227 (12.2%) |
| Free State | 209 (4.1%) | 308 (5.0%) | 517 (4.6%) | 8123 (4.4%) | 11926 (5.2%) | 20049 (4.9%) |
| Gauteng | 1884 (36.9%) | 1918 (31.2%) | 3802 (33.8%) | 51952 (28.0%) | 61629 (27.1%) | 113581 (27.5%) |
| KwaZulu-Natal | 1255 (24.6%) | 1464 (23.8%) | 2719 (24.2%) | 41449 (22.3%) | 39044 (17.2%) | 80493 (19.5%) |
| Limpopo | 63 (1.2%) | 93 (1.5%) | 156 (1.4%) | 11188 (6.0%) | 17517 (7.7%) | 28705 (6.9%) |
| Mpumalanga | 227 (4.4%) | 187 (3.0%) | 414 (3.7%) | 10140 (5.5%) | 9107 (4.0%) | 19247 (4.7%) |
| North West | 184 (3.6%) | 213 (3.5%) | 397 (3.5%) | 9073 (4.9%) | 10875 (4.8%) | 19948 (4.8%) |
| Northern Cape | 4 (0.1%) | 1 (0.0%) | 5 (0.0%) | 3251 (1.8%) | 4701 (2.1%) | 7952 (1.9%) |
| Western Cape | 514 (10.1%) | 1124 (18.3%) | 1638 (14.6%) | 25046 (13.5%) | 47913 (21.1%) | 72959 (17.7%) |

prevalence of baseline seropositivity in the immunogenicity sub-study. The median days between the last Ad26.COV2.S dose and 23 May 2022 (SHERPA start) was 188 days (IQR 171–419) among SHERPA participants and 189 days (IQR 172–406) among the non-SHERPA group (**Table 2**).

## SARS-CoV-2 infection and severe COVID-19 endpoints

As of 31 December 2022, after 3056 person-years of follow up, there were 13 recorded SARS-CoV-2 infections among SHERPA participants at median 125 days (IQR 90–154) after mRNA-1273 boosting. In contrast, 3352 infections were recorded among the non-SHERPA group after 251104 years of follow-up (**S5 Table**). There were 148 COVID-19 hospitalization or deaths among the non-SHERPA group compared to one among SHERPA participants. This immunocompromised PLWH, with a CD4 count of less than 200 cells/mm$^3$ and high

**Table 2. Recorded prior COVID-19 infections among SHERPA and non-SHERPA participants nested in the Sisonke study.**

| Characteristic | SHERPA | | | non-SHERPA | | |
|---|---|---|---|---|---|---|
| | Ad26.COV2.S + mRNA-1273 (N = 5102) | 2 Ad26.COV2.S + mRNA-1273 (N = 6146) | Total (N = 11248) | Ad26.COV2.S (N = 185676) | 2 Ad26.COV2.S (N = 227485) | Total (N = 413161) |
| Documented previous SARS- CoV-2 infection* | 398 (7.8%) | 629 (10.2%) | 1027 (9.2%) | 21480 (11.6%) | 33290 (14.6%) | 54770 (13.3%) |
| Recent prior infection# | 20 (0.4%) | 77 (1.2%) | 97 (0.9%) | 2422 (1.3%) | 6945 (3.1%) | 9367 (2.3%) |
| Non-recent prior infection+ | 378 (7.4%) | 552 (9.0%) | 930 (8.3%) | 19058 (10.3%) | 26345 (11.6%) | 45403 (11.0%). |
| **Days between last Ad26.COV2.S dose and (i) earliest date of a participant receiving the mRNA-1273 (SHERPA group) or (ii) enrolment as a comparator (non-SHERPA group)** | | | | | | |
| Median, IQR | 424 (382–443) | 172 (153–181) | 188 (171–419) | 418 (378–437) | 173 (159–182) | 189 (172–406) |

*Infection on or before 23 May 2022; Both PCR and antigen tests considered.

#Recent infection means less ≤ 90 days before enrollment for SHERPA participants and < 90 days before SHERPA study start 23 May 2022 for non-SHERPA participants.

+Non- recent infection means less > 90 days before enrollment for SHERPA participants and > 90 days before non-SHERPA participants study start 23 May 2022 for non-SHERPA participants.

HIV viremia, died of COVID-19 pneumonia with respiratory distress syndrome, six months after receiving the mRNA-1273 booster.

## Relative vaccine effectiveness of the mRNA-1273 booster

Comparing SHERPA participants to the non-SHERPA group, the unadjusted analysis showed a rVE of 59% (95%CI 29–76%) against SARS-CoV-2 infection, which did not change after adjusting for age, sex, prior SARS-CoV-2 infection, prior vaccination and geographic location (Table 3). The adjusted rVE of the booster among participants who had one prior Ad26.COV2.S dose was 77% (95%CI 9–94%) and 52% (95%CI 13–73%) for those who had two prior Ad26.COV2.S doses. In a separate model adjusting for HIV the rVE estimate only slightly changed to 57% (95%CI 25–75%). With only one severe COVID-19 case among SHERPA participants and few severe endpoints overall, the rVE estimate for COVID-19 hospitalizations and deaths lacked precision (S6 Table). In the alternative matched cohort analysis, the overall rVE against SARS-CoV-2 infection was slightly higher at 63% (95%CI 32–82%) (S7 and S8 Tables).

## Safety profile of mRNA-1273 booster

Of 11798 SHERPA participants included in the safety analysis, 228 (1.9%) participants reported 575 reactogenicity events within 7 days after the mRNA-1273 booster. Of these, 230/

**Table 3. Relative vaccine effectiveness of mRNA-1273 booster against SARS-CoV-2 infection.**

| | Unadjusted | Adjusted*: Model with comorbidities |
|---|---|---|
| | VE (95% CI) | VE (95% CI) |
| **Primary Endpoint** | | |
| SARS-CoV-2 Infection | 59% (29–76%) | 59% (29–76%) |
| **Secondary Endpoints** | | |
| SARS-CoV-2 infections after 1 previous Ad26.COV2.S | 77% (8–94%) | 77% (9–94%) |
| SARS-CoV-2 infections after 2 previous Ad26.COV2.S | 51% (13–73%) | 52% (13–73%) |

*Adjusted for age, sex, prior COVID, prior vaccination, geographical location

575 (40%) were local reactogenicity events including 97/230 (42.2%) injection site pain, 69/230 (30%) swelling, 32/230 (13.9%) induration and 32/230 (13.9%) erythema events. A total of 345 systemic reactogenicity events were reported including 123 (35.7%) malaise and/or myalgia, 120 (34.8%) headache, 66 (19.1%) fever, 17 (4.9%) chills, 10 (2.9%) nausea, 5 (1.4%) vomiting and 4 (1.2%) arthralgia. All reported local and systemic reactogenicity were mild or moderate (grade 1 or 2) and all resolved without sequalae.

Unsolicited AEs collected for 28 days after vaccination and those meeting SAE/AESI criteria collected until the end of study participation were reported by 98 (0.8%) participants, reporting a total of 115 events. Of these events, 17 events met SAE criteria. Three of the 17 SAEs also met AESI criteria, the other 4 AESIs were grade 1 and 2 anosmia and ageusia, which resolved without sequalae. No related SAEs were reported during the study.

More reactogenicity events were reported by participants with prior SARS-CoV-2 infection (3.6% vs 1.8%), HIV-negative status (2.7% vs 1.2%) and those who received two compared to one prior Ad26.COV2.S dose (2.6% vs 2.0%). In a logistic regression model adjusting for age and sex, a prior SARS-CoV-2 infection (adjusted odds ratio [aOR] = 2.03, 95%CI 1.59–2.59) and an HIV positive status (aOR = 0.49, 95%CI 0.34–0.69) were associated with local/systemic reactions, but not having received two versus one Ad26.COV2.S dose before the mRNA-1273 booster (aOR = 1.26, 95%CI 0.99–1.62), **S9 Table**.

Among 70 confirmed pregnancies, there were 39 (55.7%) full-term births, 4 (5.7%) preterm births (<37 weeks gestation), 11 (15.7%) early miscarriages (<12 weeks gestation), 7 (10.0%) stillbirths (>20 weeks gestation), and one (1.4%) ectopic pregnancy. Six (8.6%) women opted for an elective termination of pregnancy and two (2.9%) were lost to follow-up. All adverse outcomes, including miscarriages, preterm births and stillbirths were evaluated and deemed unrelated to vaccination by CRS investigators and the PSRT. Preterm births were reported in women with hypertension in pregnancy and multiple gestations, while the stillbirths (experienced by women aged 32 to 44 years) were evaluated as associated with abruptio placenta, postdate, cord around the fetal neck, and complicated home delivery (**S10 Table**).

## Immune responses after mRNA-1273 booster

Baseline neutralizing antibody titers were 2–3 fold higher in participants who had received 2 doses versus 1 dose of Ad26.COV2.S (median: 1311 vs 3269 for D614G and 508 vs 1281 for BA.4) (**Fig 2A and 2B**). The mRNA-1273 boost resulted in increased neutralizing antibodies titers for D614G after 4 weeks and then a decline by 24 weeks among those with one and two doses of Ad26.COV2.S (median baseline, 4-week and 24-week titers of 1311, 7033 and 2220 for the 1-dose group and 3269, 17653, 5526 for the 2-dose group), **Fig 2A–2C**. The antibody response after boosting was less pronounced towards the BA.4 pseudovirus despite a significant increase in rVE against this variant of concern. Baseline spike-binding antibody titers were comparable regardless of the number of previous Ad26.COV2.S doses. Binding titers significantly increased up to 6-fold 4 weeks post-boost (**Fig 2D and 2E**), and ADCC also increased significantly post-boost (**Fig 2F and 2G**). The binding and ADCC titers were lower 6 months after the boost, with similar titers observed between both Ad26.COV2.S groups (**Fig 2D–2G**). At baseline, PLWH had up to 3-fold lower binding antibody titers and up to 2-fold lower neutralizing activity against D614G compared to HIV negative participants, irrespective of prior Ad26.COV2.S doses, while ADCC levels were similar (**S4 Fig**). Antibody functions in PLWH followed similar trends after the booster as HIV-negative participants.

We examined T-cell responses in a subset of 50 participants (**Fig 3A**). At baseline, spike-specific CD4+ T-cell responses were detected in 96% of participants (**Fig 3B**). Four weeks after mRNA-1273 boosting, the frequency increased 2-fold (median: 0.074% vs 0.169%, p<0.0001),

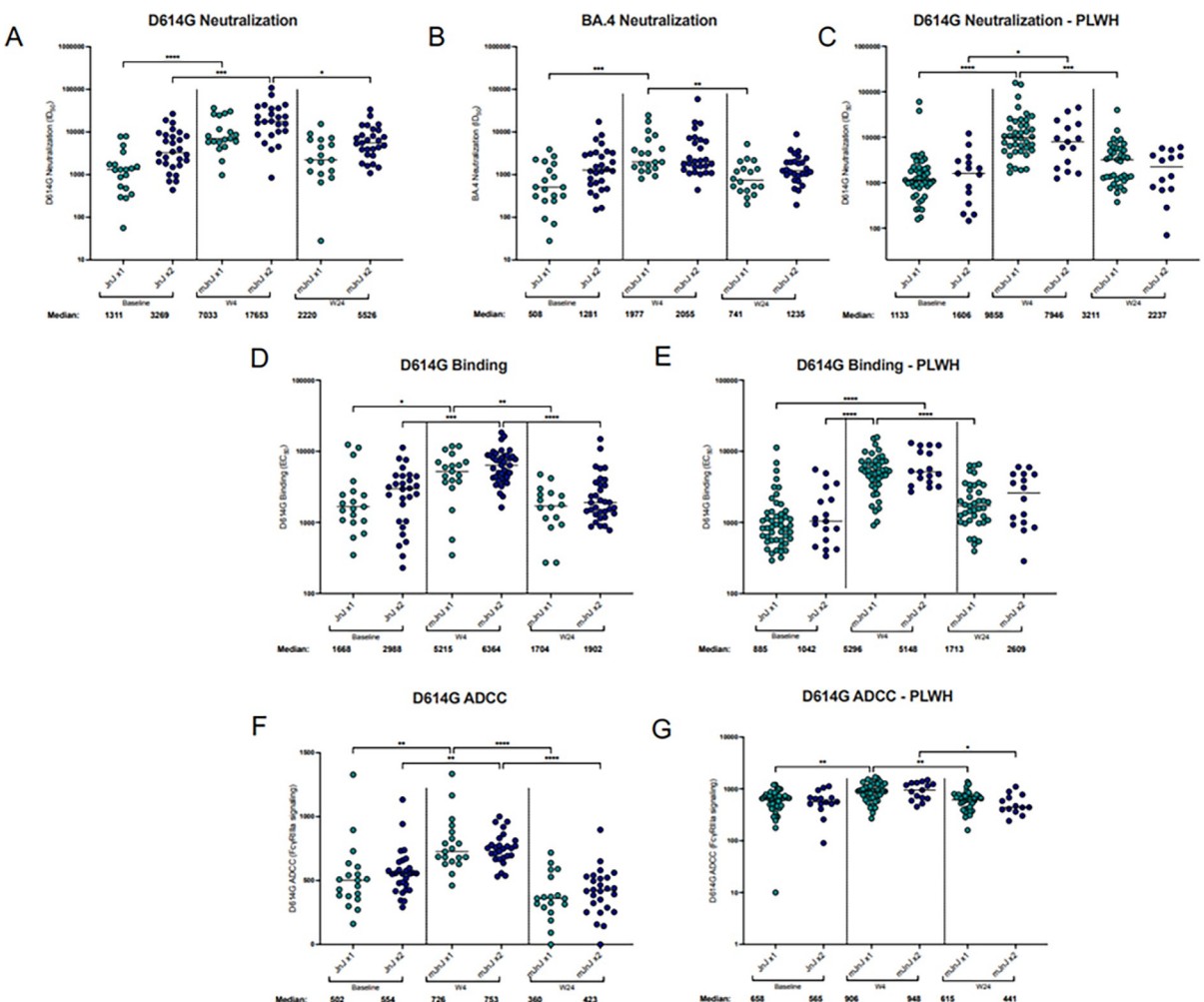

**Fig 2. Antibody response elicited after mRNA-1273 heterologous boost.** Plasma from a subset of participants were assessed to determine the antibody response elicited upon heterologous boost (mJnJ) after 1 Ad26.COV2.S dose (JnJ x1; n = 19) or 2 Ad26.COV2.S doses (JnJ x2; n = 29). The neutralization activity was measured using a SARS-CoV-2 pseudovirus-based neutralization assay with (panel A) D614G, (panel B) Omicron BA.4 and (panel C) D614G in people living with HIV (PLWH). The plasma neutralization titer is measured as an $ID_{50}$. Black horizontal bars represent medians. The threshold of detection for the neutralization assay is an $ID_{50}$ of 20. Antibody binding was measured using an in-house SARS-CoV-2 assay using the D614G full spike protein (panel D and E). An EC50 was used to measure the binding titers of the samples. ADCC activity was measured by detecting the crosslinking ability of the antibodies present in the serum (panel F and G). Relative light units were measured which correlate with the levels of FcγRIIIa signalling. For all assays, statistical significance was measured with the Kruskal-Wallis test with Dunn's multiple comparisons test. Significance is shown as: *p < 0.05, **p < 0.01, ***p < 0.001 and ****p < 0.0001. Medians and fold changes are depicted under each graph. Samples were run in duplicate for all assays. Nucleocapsid ELISA was used to identify prior SARS-CoV-2 infections. Only individuals who were seropositive at baseline were included in the analysis as there was only 1 nucleocapsid-seronegative participants in the JnJ x1 group.

followed by contraction to a median of 0.113% by 24 weeks, which was not significantly different from baseline. Spike-specific CD8+ T-cell responses occurred in 52% of participants at baseline, and followed a similar trajectory as CD4 responses, increasing significantly at 4 weeks (median: 0.034% vs 0.057%, p = 0.0029; Fig 3C) and waning to 0.047% by 24 weeks. When examining the effect of prior vaccination with Ad26.COV2.S (Fig 3D), there were significantly higher CD4+ responses at baseline in those who received two compared to one prior dose, which persisted at all timepoints after mRNA-1273 boosting. In contrast, the magnitude of CD8+ responses was similar between those who received one or two Ad26.COV2.S doses. HIV infection had no negative impact on vaccine booster responses at 4 or 24 weeks (Fig 3E).

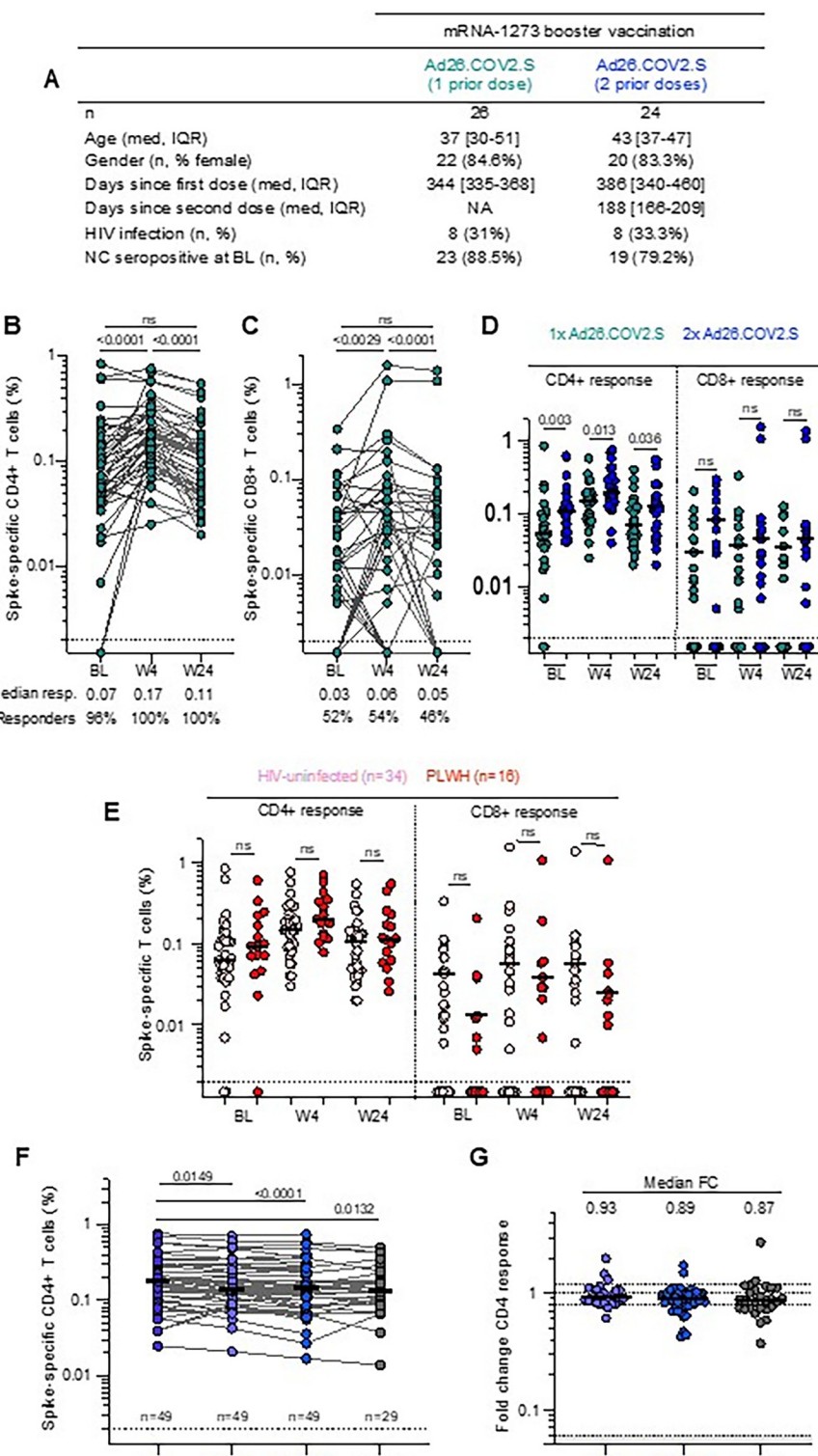

**Fig 3. T-cell response elicited after mRNA-1273 heterologous boost.** (A) Clinical characteristics of the T-cell sub-study cohort. Frequency of spike-specific CD4+ (B) and CD8+ (C) T-cells producing any of the measured cytokines (IFN-γ, IL-2 or TNF-α) in response to stimulation with a peptide pool spanning ancestral spike protein at baseline (BL), week 4 (W4) and week 24 (W24), in participants vaccinated with one or two prior doses of Ad26.COV2.S and an mRNA-1273 booster. The median response and proportion of participants with a spike-specific CD4+ or CD8+ T-cell

responses are indicated below the graphs. (D) Spike-specific CD4+ and CD8+ T-cell cytokine responses stratified according to number of Ad26.COV2.S doses received. (E) Spike CD4+ and CD8+ T-cells in HIV-uninfected participants (pink) or PLWH (red). (F) Spike-specific CD4+ T-cell responses to spike from ancestral virus and Omicron sub-lineages BA.1, BA.4/5 and XBB.1. (G) Fold change of Omicron sub-lineage responses compared to ancestral spike. Bars represent the medians. Differences between responses at different time points post boost were assessed using a two-tailed Wilcoxon paired test. Differences between participants vaccinated with one or two doses of Ad26.COV2.S and responses to different SARS-CoV-2 variants were calculated using a Mann-Whitney U test. The number of participants included in each analysis is indicated on the graphs.

When examining T-cell cross-reactivity to Omicron sub-lineages, we observed a high degree of preservation of the T-cell responses to Omicron BA.1, BA.4/5 and XBB.1 (**Fig 3F and 3G**).

## Discussion

Nested in the Sisonke phase 3b implementation study, the SHERPA trial showed that one mRNA-1273 booster administered after 1 or 2 doses of Ad26.COV2.S was safe and well tolerated regardless of HIV status, comorbidities or prior SARS-CoV-2 infection. The mRNA-1273 booster had a rVE of 59% (29–76%) in preventing SARS-CoV-2 infections overall, 77% (9–94%) after one Ad26.COV2.S dose and 52% (13–73%) after 2 Ad26.COV2.S doses up to 7 months after the boost. Breakthrough infections occurred during the Omicron wave (**S5 Fig**), thus the data confirm the benefit of heterologous mRNA-1273 boosting after priming with Ad26.COV2.S against Omicron sub-lineages.

The additional benefit of the mRNA-1273 booster was particularly high among participants who had only received a single Ad26.COV2.S previously, also indicating that the booster was especially effective among those who had received their vaccine dose more than a year previously. This finding also corroborated with strong antibody and T-cell immune responses elicited by the booster in the immunogenicity sub-study. Consistent with epidemiological data on the evolving COVID-19 pandemic, few severe COVID-19 cases were detected in the Sisonke cohort and only a single case among SHERPA participants, thereby not providing sufficient statistical power for a meaningful interpretation of rVE against COVID-19 hospitalizations and deaths.

Our findings for heterologous boosting corroborate with studies reporting on homologous mRNA-1273 boosting in the USA. In one study the rVE of a homologous mRNA-1273 boost against SARS-CoV-2 infection was 61% (95%CI 60–62%) among immunocompetent adults receiving the booster ≥150 days after priming during the Omicron wave [9]. In a second study amongst US veterans receiving a third homologous mRNA-1273 booster during the Omicron wave, the rVE was 64% (95%CI 63–65%) compared with a rVE of 12% (95%CI 10–15%) for 2-dose vaccination [10].

Earlier in the pandemic, studies assessed the VE of a heterologous prime-boost approach. One study from Sweden with 16402 participants receiving ChAdOx1 nCoV-19 prime followed by a mRNA-1273 boost found a VE of 79% (95%CI, 62–88%) against symptomatic COVID-19 infection compared with unvaccinated controls during the Delta wave [11]. An almost identical rVE was found in a study conducted across emergency departments in the USA for people who received a single Ad26.COV2.S dose followed by a mRNA-1273 boost [12].

Our findings indicate that in LMICs where vaccine availability has been limited and HIV prevalence, co-morbidities and prior SARS-CoV-2 infection are high, a heterologous mRNA-1273 boost, using ancestral vaccine after one or two Ad26.COV2.S vaccinations was protective against SARS-CoV-2 infection in healthcare workers during the Omicron wave. Our data also confirms the safety of an mRNA-1273 boost after Ad26.COV2.S priming amongst healthcare workers, including PLWH, people with comorbidities and pregnant women.

Previous epidemiological studies on pregnancy outcomes and placental studies have shown that mRNA vaccines are safe in pregnancy and that there is negligible uptake and inflammatory responses within placental tissues post vaccination [13–15]. In South Africa, only 68.3% of pregnant women have their first antenatal visit before 20 weeks gestation [16], meaning that national data on early pregnancy losses are limited. However, early pregnancy losses observed in SHERPA (15.7%) were comparable to other clinical trial cohorts (6.1–17.9%), which included early pregnancy testing and pregnancy outcome tracking [17, 18]. Stillbirths reported in this study (10.0%) were associated with polymorbid conditions, including abruptio placenta, cord round the fetal neck and postdate, that provided clear alternative causes to the occurrence of the events. While the stillbirth rate was higher than in two clinical trial cohorts among younger, HIV negative women in HIV vaccine trials (1.5–3.5%), our data are in keeping with a 10% increase in stillbirths noted nationally in recent review periods [19], as well as the generally higher stillbirth rates reported among older cohorts and cohorts of PLWH [20–22].

Antibody responses were higher among participants with two compared to one prior Ad26. COV2.S dose and were boosted with a heterologous mRNA vaccine among both groups, as reported previously [2, 23, 24]. Despite some evidence that Ad26.COV2.S may lead to slower waning of immune responses than mRNA vaccines [25, 26], the decline of antibody responses at 6 months in SHERPA suggests that the Ad26.COV2.S as a primary dose(s) may not significantly increase the durability of mRNA-boosted antibody responses. Although PLWH had lower antibody responses at baseline, mRNA-1273 boosting resulted in comparable titers to HIV-negative participants. mRNA-1273 boosting increased spike-specific T-cell responses, as described previously for heterologous vaccination regimens [2, 27]. These responses were durable and highly cross-reactive to Omicron sub-lineages.

SHERPA had some limitations. First, it was a single arm trial nested in the larger Sisonke trial, and participants were not randomized. As in Sisonke, we relied on routinely collected data to ascertain COVID-19 endpoints. However, this applied to both mRNA-1273 boosted and unboosted study groups, therefore reducing the potential for reporting bias. Second, we observed too few severe COVID-19 endpoints to allow for a well-powered assessment of rVE against severe disease. Contributing factors could have been that SHERPA was conducted during the Omicron wave, when generally less severe disease was reported [28], South Africa has a high SARS-CoV-2 seroprevalence of up to 80% [29], and all participants had at least received one dose of Ad26.COV2.S previously. Third, we presented short to medium term rVE results, and were not able to assess the long-term protection of the mRNA-1273 booster due to national changes in COVID-19 case reporting. Fourth, while the adjusted cohort and matched cohort approaches seek to emulate a clinical trial from observational data, residual bias emanating from the impact of unmeasured confounders or selection bias may still remain and therefore results should be carefully interpreted. Finally, we were unable to compare pregnancy outcomes in the SHERPA study with the larger Sisonke trial population, because pregnant women were initially not eligible for Sisonke, and pregnancy outcomes were not systematically collected.

Our data show that a heterologous mRNA-1273 boost following one or two Ad26.COV2.S priming among healthcare workers including those with HIV, comorbidities and pregnancy can be included in national policy. We also demonstrate that boosting with the ancestral mRNA-1273 vaccine is still effective despite the evolution of SARS-CoV-2 into Omicron sub-lineage. This has policy and programming complications as homologous boosting and variant vaccines may not be accessible in these settings. In conclusion, the SHERPA trial showed a high rVE of the mRNA-1273 booster during the Omicron wave among healthcare workers who had received one or two previous doses of Ad26.COV2.S. The heterologous booster was safe and well tolerated, immunogenic and effective in a LMIC with high HIV prevalence. This

study provides in principle support for transitioning from homologous to heterologous boosters with Ad26 and mRNA platforms in the event of future pandemics where vaccine availability may be limited. Additionally, it provides evidence that variant viruses do not always need variant vaccines. While this is especially pertinent to LMIC settings, it does not preclude the need to monitor variants and the neutralisation capacity of parent vaccines before pandemic vaccination policies are finalised and implemented. There is a need for future research to ensure continued monitoring of vaccine durability and effectiveness against evolving variants.

## Supporting information

**S1 Checklist. CONSORT checklist.**
(DOC)

**S1 Table. Study schema of Sisonke mRNA-1273 boost and comparator groups.**
(DOCX)

**S2 Table. Eligibility criteria for the SHERPA trial.**
(DOCX)

**S3 Table. Screening and enrolment visit procedures in the SHERPA study.**
(DOCX)

**S4 Table. Schedule of evaluation for participants in the safety and immunogenicity sub-study.**
(DOCX)

**S5 Table. Number of SARS-CoV-2 infection and severe Covid-19 events and total exposure time (in years) among SHERPA and non-SHERPA Sisonke participants.**
(DOCX)

**S6 Table. Relative vaccine effectiveness of the mRNA-1273 booster against COVID-19 hospitalizations or death estimates.**
(DOCX)

**S7 Table. Matched cohort analysis matching SHERPA and non-SHERPA participants 1:1 on key variables.**
(DOCX)

**S8 Table. Relative vaccine effectiveness of the mRNA-1273 booster using the matched cohort analysis approach.**
(DOCX)

**S9 Table. Multivariable logistic regression model of local/systemic reactions adjusted for age and sex.**
(DOCX)

**S10 Table. Adverse pregnancy outcomes among pregnant women.**
(DOCX)

**S1 Fig. Titer comparison between SARS-CoV-2 lentivirus-based and VSV-based pseudovirus neutralization assays.** Fifty-one samples from HIV-uninfected individuals were tested in both the SARS-CoV-2 lentivirus-based pseudovirus assay and the SARS-CoV-2 VSV-based neutralization assay. Samples were chosen to represent low, middle and high titer values. Titers are depicted as ID50 values for both assays. The correlation between the two assays was

measured using the Spearman's correlation in Graphpad Prism v10.0.2.
(DOCX)

**S2 Fig. Safety and COVID-19 surveillance in the SHERPA study.**
(DOCX)

**S3 Fig. Flow diagram showing the adjusted cohort study design.**
(DOCX)

**S4 Fig. Comparison of antibody response elicited after mRNA-1273 heterologous boost in HIV-negative participants and PLWH.** The binding (Panel A and D), neutralization (Panel B and C) and ADCC (Panel C and F) were measured in individuals who were HIV-negative and people living with HIV (PLWH). Comparisons were separated based on number of Ad26. COV2.S with 1 doses Ad26.COV2.S (Panels A-C) and 2 doses of Ad26.COV2.S (Panels D-F). The plasma neutralization titer is measured as an $ID_{50}$. Black horizontal bars represent medians. The threshold of detection for the neutralization assay is an $ID_{50}$ of 20. Antibody binding was measured using an in-house SARS-CoV-2 assay using the D614G full spike protein. An EC50 was used to measure the binding titers of the samples. ADCC activity was measured by detecting the crosslinking ability of the antibodies present in the serum. Relative light units were measured which correlate with the levels of FcγRIIIa signalling. For all assays, statistical significance was measured with the Kruskal-Wallis test with Dunn's multiple comparisons test. Significance is shown as: $^*p < 0.05$, $^{**}p < 0.01$, $^{***}p < 0.001$ and $^{****}p < 0.0001$. Medians and fold changes are depicted under each graph. Samples were run in duplicate for all assays.
(DOCX)

**S5 Fig. SARS-CoV-2 infections and circulating viral strains in South Africa during SHERPA trial 2022–2023 (n = 15 392\*).**
(DOCX)

**S1 Protocol. Study protocol.**
(PDF)

**S1 Text.**
(DOCX)

## Acknowledgments

We thank the participants, the protocol and safety teams, the site-Principal Investigators, and their teams, the laboratory teams, SAHPRA, and the ethics committees for their contribution and oversight of the study.

## Author Contributions

**Conceptualization:** Nigel Garrett, Tarylee Reddy, Nonhlanhla Yende-Zuma, Azwidhwi Takalani, Penny L. Moore, Brett Leav, Linda-Gail Bekker, Glenda E. Gray, Ameena Goga.

**Data curation:** Nigel Garrett, Azwidhwi Takalani, Annie Bodenstein, Phumeza Jonas, Imke Engelbrecht, Waasila Jassat, Harry Moultrie, Ishen Seocharan, Simone I. Richardson, Millicent A. Omondi, Rofhiwa Nesamari, Roanne S. Keeton, Catherine Riou, Thandeka Moyo-Gwete, Craig Innes, Zwelethu Zwane, Kathy Mngadi, William Brumskine, Nivashnee Naicker, Disebo Potloane, Sharlaa Badal-Faesen, Steve Innes, Shaun Barnabas, Johan Lombaard, Katherine Gill, Maphoshane Nchabeleng, Elizma Snyman, Friedrich Petrick,

Elizabeth Spooner, Logashvari Naidoo, Dishiki Kalonji, Vimla Naicker, Nishanta Singh, Rebone Maboa, Pamela Mda, Daniel Malan, Anusha Nana, Mookho Malahleha, Philip Kotze, Jon J. Allagappen, Andreas H. Diacon, Gertruida M. Kruger, Faeezah Patel, Penny L. Moore, Wendy A. Burgers, Ameena Goga.

**Formal analysis:** Nigel Garrett, Tarylee Reddy, Nonhlanhla Yende-Zuma, Azwidhwi Takalani, Annie Bodenstein, Harry Moultrie, Ishen Seocharan, Simone I. Richardson, Millicent A. Omondi, Rofhiwa Nesamari, Roanne S. Keeton, Catherine Riou, Thandeka Moyo-Gwete, Penny L. Moore, Wendy A. Burgers, Glenda E. Gray, Ameena Goga.

**Funding acquisition:** Nigel Garrett, Kubashni Woeber, Linda-Gail Bekker, Glenda E. Gray, Ameena Goga.

**Investigation:** Nigel Garrett, Tarylee Reddy, Nonhlanhla Yende-Zuma, Azwidhwi Takalani, Kubashni Woeber, Phumeza Jonas, Imke Engelbrecht, Waasila Jassat, Ishen Seocharan, Jackline Odhiambo, Kentse Khuto, Simone I. Richardson, Millicent A. Omondi, Rofhiwa Nesamari, Roanne S. Keeton, Catherine Riou, Thandeka Moyo-Gwete, Craig Innes, Zwe-lethu Zwane, Kathy Mngadi, William Brumskine, Nivashnee Naicker, Disebo Potloane, Sharlaa Badal-Faesen, Steve Innes, Shaun Barnabas, Johan Lombaard, Katherine Gill, Maphoshane Nchabeleng, Elizma Snyman, Friedrich Petrick, Elizabeth Spooner, Logash-vari Naidoo, Dishiki Kalonji, Vimla Naicker, Nishanta Singh, Rebone Maboa, Pamela Mda, Daniel Malan, Anusha Nana, Mookho Malahleha, Philip Kotze, Jon J. Allagappen, Andreas H. Diacon, Gertruida M. Kruger, Faeezah Patel, Penny L. Moore, Wendy A. Burgers, Linda-Gail Bekker, Glenda E. Gray, Ameena Goga.

**Methodology:** Nigel Garrett, Tarylee Reddy, Nonhlanhla Yende-Zuma, Azwidhwi Takalani, Waasila Jassat, Harry Moultrie, Debbie Bradshaw, Ishen Seocharan, Jackline Odhiambo, Wendy A. Burgers, Linda-Gail Bekker, Ameena Goga.

**Project administration:** Nigel Garrett, Nonhlanhla Yende-Zuma, Azwidhwi Takalani, Kubashni Woeber, Annie Bodenstein, Phumeza Jonas, Imke Engelbrecht, Ishen Seocharan, Jackline Odhiambo, Kentse Khuto, Simone I. Richardson, Roanne S. Keeton, Catherine Riou, Thandeka Moyo-Gwete, Penny L. Moore, Kate Anteyi, Brett Leav, Linda-Gail Bekker, Glenda E. Gray, Ameena Goga.

**Resources:** Linda-Gail Bekker, Glenda E. Gray.

**Software:** Ishen Seocharan.

**Supervision:** Nigel Garrett, Tarylee Reddy, Nonhlanhla Yende-Zuma, Azwidhwi Takalani, Kubashni Woeber, Ishen Seocharan, Kentse Khuto, Penny L. Moore, Wendy A. Burgers, Linda-Gail Bekker, Glenda E. Gray, Ameena Goga.

**Validation:** Nigel Garrett, Tarylee Reddy, Nonhlanhla Yende-Zuma, Azwidhwi Takalani, Ishen Seocharan, Rofhiwa Nesamari, Wendy A. Burgers, Glenda E. Gray, Ameena Goga.

**Visualization:** Nigel Garrett, Ishen Seocharan, Simone I. Richardson.

**Writing – original draft:** Nigel Garrett, Ameena Goga.

**Writing – review & editing:** Nigel Garrett, Tarylee Reddy, Nonhlanhla Yende-Zuma, Azwidhwi Takalani, Kubashni Woeber, Phumeza Jonas, Imke Engelbrecht, Waasila Jassat, Harry Moultrie, Debbie Bradshaw, Ishen Seocharan, Jackline Odhiambo, Kentse Khuto, Simone I. Richardson, Millicent A. Omondi, Rofhiwa Nesamari, Roanne S. Keeton, Cather-ine Riou, Thandeka Moyo-Gwete, Craig Innes, Zwelethu Zwane, Kathy Mngadi, William Brumskine, Nivashnee Naicker, Disebo Potloane, Sharlaa Badal-Faesen, Steve Innes, Shaun

Barnabas, Johan Lombaard, Katherine Gill, Maphoshane Nchabeleng, Elizma Snyman, Friedrich Petrick, Elizabeth Spooner, Logashvari Naidoo, Dishiki Kalonji, Vimla Naicker, Nishanta Singh, Rebone Maboa, Pamela Mda, Daniel Malan, Anusha Nana, Mookho Malahleha, Philip Kotze, Jon J. Allagappen, Andreas H. Diacon, Gertruida M. Kruger, Faeezah Patel, Penny L. Moore, Wendy A. Burgers, Kate Anteyi, Brett Leav, Linda-Gail Bekker, Glenda E. Gray, Ameena Goga.

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
