## [Decision Letter · Decision Letter 0]

8 Jul 2024

PGPH-D-24-00923

Safety, Effectiveness and Immunogenicity of heterologous mRNA-1273 Boost after Prime with Ad26.COV2.S among Healthcare Workers in South Africa: the single-arm, open-label, Phase 3 SHERPA Study

Dear Dr. Garrett,

Thank you for submitting your manuscript to PLOS Global Public Health. After careful consideration, we feel that it has merit but does not fully meet PLOS Global Public Health’s publication criteria as it currently stands. Therefore, we invite you to submit a revised version of the manuscript that addresses the points raised during the review process.

We look forward to receiving your revised manuscript.

Kind regards,

Abram L. Wagner, PhD, MPH

Academic Editor

Journal Requirements:

2. We have noticed that you have uploaded Supporting Information files, but you have not included a list of legends. Please add a full list of legends for your Supporting Information files after the references list.

3. In the online submission form, you indicated that "Considering this is a clinical trial, universal access to data may not be possible to protect clinical trial participants. However, SHERPA data will be made available by the authors upon reasonable request". 

3. Uploaded as supplementary information.

Additional Editor Comments (if provided):

Reviewers' comments:

Reviewer's Responses to Questions

**Comments to the Author**

1. Does this manuscript meet PLOS Global Public Health’s publication criteria? Is the manuscript technically sound, and do the data support the conclusions? The manuscript must describe methodologically and ethically rigorous research with conclusions that are appropriately drawn based on the data presented.

Reviewer #1: Yes

Reviewer #2: Yes

Reviewer #3: Yes

Reviewer #4: Yes

2. Has the statistical analysis been performed appropriately and rigorously?

Reviewer #1: Yes

Reviewer #2: Yes

Reviewer #3: Yes

Reviewer #4: Yes

3. Have the authors made all data underlying the findings in their manuscript fully available (please refer to the Data Availability Statement at the start of the manuscript PDF file)?

Reviewer #1: Yes

Reviewer #2: Yes

Reviewer #3: Yes

Reviewer #4: Yes

4. Is the manuscript presented in an intelligible fashion and written in standard English?

Reviewer #1: Yes

Reviewer #2: Yes

Reviewer #3: Yes

Reviewer #4: Yes

5. Review Comments to the Author

Reviewer #1: The followings are the observation:

1. The participants of the study have been selected based on few criteria. What are those criteria?

2. The participants immune status was not assessed. Please Explain why?

3. The ELISA was applied to detect Ig status. Does the ELISA kit validated?

4. The process of antigen preparation is not mentioned here. Please write in details.

5. The exposure history of exposure is not mentioned here. Please describe this.

6. How did the authors measure the T-Cell count? This is missing in the methodology section.

7. The results section is missing the AEs or SAEs. Please mention.

Reviewer #2: The manuscript is generally well written except for a few grammatical errors

1.The abstract and introduction are difficult to follow. The authors need to make it understandable to the reader,

2. Abbrevaitions should be defined should be defined when first used e.g mcg

Reviewer #3: Garrett et al. present extensive observational data assessing the Safety, Effectiveness, and Immunogenicity of heterologous mRNA-1273 Boost following Ad26.COV2.S Prime among Healthcare Workers in South Africa: the single-arm, open-label, Phase 3 SHERPA Study. While the manuscript reveals significant results which is very important for health. Significant revisions and some issues needs to be addressed as highlighted below for the manuscript to meet publishable standards.

Title:

The title effectively communicates the scope, context, and key outcomes of the study however, the title is quite lengthy, which may make it harder for readers to quickly grasp the main focus of the study

Authors: Why too many authors (35)?

Abstract:

The abstract is well written, though some issues should be addressed

In line 71 SHERPA should be written full in its first appearance

In line 97 what are the reactogenicity events the authors are referring to? This should be stated

In line 100 the authors should be specific by mentioning the specific antibodies and immune responses that was observed, generalization cannot do justice here.

Introduction:

In line 111 instead of stating 2 years let the authors state down specific years (e.g 2020-2021) to bring more clarity

Materials and Methods

In line 183 the immediate reactogenicity events should be mentioned

In line 190-201 the authors should describe briefly the procedures for the assays mentioned.

The ethical approval number should be stated by authors

Was sample size determination considered in this study considering the fact that the study involved 501230 Sisonke participants? The authors should shade light on this.

Results:

In line 292-293. The author states that “The majority (79.3%) of SHERPA participants were female and median age was 41 years (IQR 35–48)”. However, the 20.7% is not mentioned including the median age. These results for men should be equally described.

In line 408 “ADCC had smaller but significant increases post-boost”. This statement is confusing it is either there is statistical significance or not…but having smaller and significant in the same sentence is misleading. The authors should revise this statement

In Line 453-457 the sentence is too long and redundant, thus the authors should revise it to bring clarity

The use of the Ad26.COV2.S vaccine, is linked with thrombosis and thrombocytopenia syndrome, the authors should at least provide some results that touches in any of these two

Discussion:

The discussion section effectively synthesizes the study's results, placing them within the context of current knowledge. However, the authors should address the implications for practice, policy, and future research.

Whereas it provides a balanced interpretation of the findings the authors have failed to acknowledge the limitations and thus this study has not clearly suggested directions for further investigation.

Nevertheless, the overall, the discussion contributes valuable insights into the evolving landscape of COVID-19 vaccination strategies, particularly in diverse and high-risk populations.

Reviewer #4: The study design leveraging data from the SHERPA trial nested within the Sisonke study provides robust evidence on the effectiveness and safety of the mRNA-1273 booster in individuals previously vaccinated with Ad26.COV2.S.

The large sample size and comprehensive data collection allow for detailed subgroup analyses, particularly regarding different doses of Ad26.COV2.S and participant demographics (e.g., HIV status).

The inclusion of both unadjusted and adjusted analyses for vaccine effectiveness against SARS-CoV-2 infections adds depth to the findings, enhancing the reliability of the reported results.

Limitations

While the study provides valuable insights into the effectiveness of the mRNA-1273 booster, there are limitations in the follow-up period, especially regarding the duration of immune response post-booster. Longer-term data would strengthen the assessment of sustained protection.

The exclusion criteria, such as participants receiving a third Ad26.COV2.S or BNT162b2 booster, could potentially influence the generalizability of the findings, particularly in settings where multiple boosters are administered.

The interpretation of vaccine effectiveness against severe COVID-19 outcomes is limited by the small number of events, particularly in the SHERPA cohort, which affects the precision of these estimates.

Recommendations

Provide additional discussion on the implications of the findings in the context of emerging variants, especially Omicron, given the breakthrough infections observed during this period.

Consider discussing the potential implications of these findings on vaccine policy and public health strategies, particularly in populations with varying levels of prior exposure to COVID-19 and different vaccine schedules.

Strengthen the conclusions by addressing the limitations related to data completeness, particularly in subgroup analyses, and provide insights into future research directions, such as the need for continued monitoring of vaccine durability and effectiveness against evolving variants.

Overall Impact

This study significantly contributes to the evolving literature on COVID-19 vaccine effectiveness and safety, particularly in the context of heterologous booster strategies.

The findings are timely and relevant for informing vaccination strategies globally, especially in populations with initial Ad26.COV2.S vaccinations, including considerations for special populations such as individuals living with HIV.

6. PLOS authors have the option to publish the peer review history of their article (what does this mean?). If published, this will include your full peer review and any attached files.

**Do you want your identity to be public for this peer review?** For information about this choice, including consent withdrawal, please see our Privacy Policy.

Reviewer #1: **Yes: **Dr. Md. Abdullah Yusuf

Reviewer #2: **Yes: **ESTER ACEN

Reviewer #3: **Yes: **Dr. JAMES NYABUGA NYARIKI

Reviewer #4: No

---

## [Decision Letter · Decision Letter 1]

30 Sep 2024

PGPH-D-24-00923R1

Safety, Effectiveness and Immunogenicity of heterologous mRNA-1273 Boost after Prime with Ad26.COV2.S among Healthcare Workers in South Africa: the single-arm, open-label, Phase 3 SHERPA Study

Dear Dr. Garrett,

Thank you for submitting your manuscript to PLOS Global Public Health. After careful consideration, we feel that it has merit but does not fully meet PLOS Global Public Health’s publication criteria as it currently stands. Therefore, we invite you to submit a revised version of the manuscript that addresses the points raised during the review process.

We look forward to receiving your revised manuscript.

Kind regards,

Abram L. Wagner, PhD, MPH

Academic Editor

Journal Requirements:

Additional Editor Comments (if provided):

Below is our review from a statistical expert, which is required as part of publication of any paper reporting a clinical trial.

Reviewers' comments:

Reviewer's Responses to Questions

**Comments to the Author**

1. If the authors have adequately addressed your comments raised in a previous round of review and you feel that this manuscript is now acceptable for publication, you may indicate that here to bypass the “Comments to the Author” section, enter your conflict of interest statement in the “Confidential to Editor” section, and submit your "Accept" recommendation.

Reviewer #5: (No Response)

2. Does this manuscript meet PLOS Global Public Health’s publication criteria? Is the manuscript technically sound, and do the data support the conclusions? The manuscript must describe methodologically and ethically rigorous research with conclusions that are appropriately drawn based on the data presented.

Reviewer #5: Yes

3. Has the statistical analysis been performed appropriately and rigorously?

Reviewer #5: No

4. Have the authors made all data underlying the findings in their manuscript fully available (please refer to the Data Availability Statement at the start of the manuscript PDF file)?

Reviewer #5: Yes

5. Is the manuscript presented in an intelligible fashion and written in standard English?

Reviewer #5: Yes

6. Review Comments to the Author

Reviewer #5: The study investigates the safety and effectiveness of a heterologous mRNA-1273 booster in healthcare workers primed with Ad26.COV2.S, making it a clear piece of primary research. The methodology is well-defined with the use of a single-arm, open-label, phase 3 design nested within a larger trial (Sisonke).

The are some comments as listed below:

1. The lack of randomization is the most significant limitation when comparing the SHERPA study to the unboosted Sisonke cohort. Randomization is critical for ensuring that treatment groups are comparable in terms of both observed and unobserved factors, thereby minimizing confounding and selection bias. Since SHERPA lacks randomization, the comparison with the unboosted Sisonke participants comes with several inherent risks.

- Selection Bias: Without randomization, the SHERPA participants (those who received the mRNA-1273 booster) may differ systematically from the unboosted Sisonke participants in ways that are not controlled for. For instance, participants who opted to receive the booster might be more health-conscious or have different risk factors, comorbidities, or exposure risks than those who did not get boosted.

- Even though statistical adjustments (e.g., controlling for age, sex, prior infection) are used in the Cox model, unmeasured confounders (e.g., occupation type, risk behaviors, healthcare access) could bias the results. These factors would likely have been balanced if randomization were used.

- Confounding Variables: In non-randomized studies, confounding variables—both measured and unmeasured—can skew the results. The SHERPA study adjusts for several confounders like prior SARS-CoV-2 infection, comorbidities, and geographic location, but the absence of randomization leaves room for residual confounding.

For example, SHERPA participants could have a different exposure to the virus compared to unboosted Sisonke participants, especially if the timing of vaccination and circulating variants differed. Such confounding variables may not be adequately captured by the existing adjustments.

- Immortal Time Bias: Immortal time bias can arise when participants in one group (e.g., SHERPA) are considered immune to the outcome (infection, severe disease) for a certain period. In this case, the time-varying nature of the mRNA-1273 booster is accounted for in the model, but without randomization, there’s still a risk that some bias persists, particularly in how the time frames for receiving the vaccine differ between groups.

- Baseline Differences: Randomization ensures that groups start with comparable baseline characteristics. In SHERPA’s comparison with Sisonke, the lack of randomization means there could be significant differences at baseline, such as the time since initial vaccination or underlying health conditions, which could affect the vaccine effectiveness estimates.

2. Statistical Techniques Attempt to Mitigate But Not Replace Randomization:

- Cox Proportional Hazards with Time-Varying Covariates and Matched Cohort Analyses are used in the study to attempt to balance the groups and reduce bias. While these methods can help control for measured confounders, they cannot account for unmeasured variables, leaving the comparison vulnerable to bias.

- Techniques like propensity score matching (PSM), inverse probability of treatment weighting (IPTW), or stratification based on propensity scores can improve the balance of measured confounders but still fall short of the robustness that randomization would provide. These methods adjust for known covariates but cannot guarantee that all relevant confounders are accounted for.

The above points in 1) and 2) need to be addressed. The lack of randomization is a critical limitation that cannot be fully addressed through statistical adjustments alone. While techniques like Cox models and propensity score matching can help to balance observed covariates, they cannot compensate for unmeasured confounders and selection bias. As a result, the comparison between SHERPA and the Sisonke cohort must be interpreted with caution. The absence of randomization reduces the strength of the causal claims and introduces uncertainty about whether the observed differences in outcomes are truly due to the mRNA-1273 booster.

For a more robust and reliable comparison, a randomized controlled trial (RCT) or at least a well-designed randomized arm within the SHERPA study would have been ideal.

7. PLOS authors have the option to publish the peer review history of their article (what does this mean?). If published, this will include your full peer review and any attached files.

**Do you want your identity to be public for this peer review?** For information about this choice, including consent withdrawal, please see our Privacy Policy.

Reviewer #5: No

---

## [Editor Report · Decision Letter 2]

25 Oct 2024

Safety, Effectiveness and Immunogenicity of heterologous mRNA-1273 Boost after Prime with Ad26.COV2.S among Healthcare Workers in South Africa: the single-arm, open-label, Phase 3 SHERPA Study

PGPH-D-24-00923R2

Dear Prof Garrett,

We are pleased to inform you that your manuscript 'Safety, Effectiveness and Immunogenicity of heterologous mRNA-1273 Boost after Prime with Ad26.COV2.S among Healthcare Workers in South Africa: the single-arm, open-label, Phase 3 SHERPA Study' has been provisionally accepted for publication in PLOS Global Public Health.

Best regards,

Abram L. Wagner, PhD, MPH

Academic Editor